# Synthetic Retinoid Seletinoid G Improves Skin Barrier Function through Wound Healing and Collagen Realignment in Human Skin Equivalents

**DOI:** 10.3390/ijms21093198

**Published:** 2020-04-30

**Authors:** Eun-Soo Lee, Yujin Ahn, Il-Hong Bae, Daejin Min, Nok Hyun Park, Woonggyu Jung, Se-Hwa Kim, Yong Deog Hong, Won Seok Park, Chang Seok Lee

**Affiliations:** 1Amorepacific Corporation R&D Center, Yongin 17074, Gyunggi-do, Korea; soopian82@gmail.com (E.-S.L.); baelong98@naver.com (I.-H.B.); djmin@amorepacific.com (D.M.); aquareve@amorepacific.com (N.H.P.); hydhong@amorepacific.com (Y.D.H.); wspark@amorepacific.com (W.S.P.); 2Department of Biomedical Engineering, Ulsan National Institute of Science and Technology (UNIST), Ulsan 44919, Korea; whi506@naver.com (Y.A.); wgjung@unist.ac.kr (W.J.); 3Center for Nano-Bio Measurement, Korea Research Institute of Standards and Science, Daejeon 34113, Korea; shkim@kriss.re.kr; 4Department of Beauty and Cosmetic Science, Eulji University, Seongnam 13135, Gyunggi-do, Korea

**Keywords:** seletinoid G, wound healing, keratinocyte, human skin equivalents, optical coherence tomography, second harmonic generation

## Abstract

The outer epidermal skin is a primary barrier that protects the body from extrinsic factors, such as ultraviolet (UV) radiation, chemicals and pollutants. The complete epithelialization of a wound by keratinocytes is essential for restoring the barrier function of the skin. However, age-related alterations predispose the elderly to impaired wound healing. Therefore, wound-healing efficacy could be also considered as a potent function of an anti-aging reagent. Here, we examine the epidermal wound-healing efficacy of the fourth-generation retinoid, seletinoid G, using HaCaT keratinocytes and skin tissues. We found that seletinoid G promoted the proliferation and migration of keratinocytes in scratch assays and time-lapse imaging. It also increased the gene expression levels of several keratinocyte proliferation-regulating factors. In human skin equivalents, seletinoid G accelerated epidermal wound closure, as assessed using optical coherence tomography (OCT) imaging. Moreover, second harmonic generation (SHG) imaging revealed that seletinoid G recovered the reduced dermal collagen deposition seen in ultraviolet B (UVB)-irradiated human skin equivalents. Taken together, these results indicate that seletinoid G protects the skin barrier by accelerating wound healing in the epidermis and by repairing collagen deficiency in the dermis. Thus, seletinoid G could be a potent anti-aging agent for protecting the skin barrier.

## 1. Introduction

The skin, which comprises the outer epidermis, underlying connective tissue and the dermis, functions as a barrier that protects the body from environmental stressors, such as pathogens and physical stress [1]. Skin injuries may damage the epidermis and dermal layers, necessitating repair through wound-healing processes, such as proliferation and re-epithelialization [2]. When this well-ordered process is disrupted, as can be seen with age, the injury may develop into a chronic wound [3,4]. Aged skin is more prone to injury, and scars formed by aged skin are weaker than those formed by younger skin. Studies on the relationship between aging and wound-healing processes have indicated that the altered mechanical environment of aged skin may largely account for age-related delays in healing [5,6,7].

The term “retinoids” refers to the naturally occurring or synthetic members, precursors or derivatives of vitamin A that bind to nuclear retinoid receptors, such as retinoid X receptors (RXRs) and retinoic acid receptors (RARs). Their key functions in physiology are controlling cellular proliferation and differentiation. In skin, retinoids promote keratinocyte proliferation, strengthen the protective function of the epidermis, protect collagen against degradation and inhibit metalloproteinase activity; thus, they have an anti-aging effect on skin. Topical retinoids have been shown to prevent and repair clinical features of both intrinsic aging and photo-aging [8,9,10,11]. 

In 2005, our institute developed seletinoid G, a novel pyranone derivative, 2-((3E)-4(2H,3H-benzo[3,4-d]1,3-dioxolan-5-yl)-2-oxo-but-3-enyloxy)-5-hydroxy-4H-pyran-4-one, as a novel synthetic retinoid [12]. It was synthesized by sequential reaction of kojic acid (KA) with thionyl chloride and then with 3,4-(methylenedioxy) cinnamic acid (CA). Seletinoid G represents a fourth-generation retinoid that treats intrinsic/photo-aging, because topical application of seletinoid G under occlusion induced no skin irritation; this was in contrast to tretinoin, which caused severe erythema [13]. Studies have shown that seletinoid G repairs altered connective tissue in old skin [12], promotes adiponectin production during adipogenesis [14], stimulates adiponectin-induced hair growth factors in human dermal papilla cells [15], dually modulates peroxisome proliferator-activated receptor α/γ (PPARα/γ) and inhibits ultraviolet B (UVB) irradiation-induced inflammation in human epidermal keratinocytes [16]. However, there are no reports about the wound-healing efficacy of seletinoid G for anti-aging.

In the current study, we used several advanced protocols to examine the wound-healing efficacy of seletinoid G. Scratch assays and automated time-lapse imaging were used to visualize the real-time migration of HaCaT cells (an immortalized normal human keratinocytes). Optical coherence tomography (OCT) was applied to examine wound healing in the epidermis layer of human skin equivalents treated with and without seletinoid G. Finally, three-dimensional investigation using second harmonic generation (SHG) imaging was used to reveal that seletinoid G realigns collagen deposition in the dermis of human skin equivalents.

## 2. Results

### 2.1. Cell Viability and Proliferation in Seletinoid G-Treated HaCaT Cells and Normal Human Dermal Fibroblasts (NHDF)

Prior to observing the wound-healing efficacy of seletinoid G, we defined the concentration of seletinoid G that was tolerated in the context of the cell viability and proliferation of human keratinocytes (HaCaT cells) and primary normal human dermal fibroblasts (NHDF). We applied various concentrations of seletinoid G to HaCaT cells and NHDF for 24 or 48 h. As shown in Figure 1, seletinoid G treatment at concentrations up to 25 µM had no effect on cell viability at 24 h and clearly increased the number of HaCaT cells up to 48 h. Similar results were obtained in NHDF (Figure 1). Thus, we tested seletinoid G at concentrations up to 25 µM in our in vitro experiments.

### 2.2. Wound-Healing Efficacy of Seletinoid G

Next, we tested the wound-healing efficacy of seletinoid G in HaCaT cells. HaCaT cells were seeded to culture plates and grown for 24 h, and the formed monolayers were scratched in a straight line on the plate. Wound healing was then monitored for 48 h in the presence of various concentrations of seletinoid G. As shown in Figure 2A and the Appendix A, we observed that seletinoid G-treated cells were more proliferative and/or migrated more to close the wounded area, compared to the control cells. When we measured the wound-healing area every hour for 48 h using time-lapse imaging microscopy, we observed that HaCaT cells treated with 6, 12 and 25 µM seletinoid G all covered the wounded area significantly better than the control cells, although 12 µM seletinoid G tended to be more effective than 25 µM seletinoid G (Figure 2B).

To prepare the wounded skin-equivalent model, a full-thickness skin model comprising human epidermal keratinocytes in an epidermis and human dermal fibroblasts in a dermis was wounded with a 3-mm biopsy punch and topically treated with seletinoid G (12 and 25 µM) every other day. After 3 or 6 days, the re-epithelialized areas were three-dimensionally measured using optical coherence tomography (OCT) and H&E staining. As shown in Figure 3, we observed that topical treatment with 12 μM seletinoid G dramatically accelerated wound closure (blue colors) compared to the control group at day 3. As seen in Table 1, 12 μM seletinoid G significantly showed a wound-healing area of 84.7% while the control group showed a wound-healing area of 51.3%. In the case of 25 μM seletinoid G, there was a tendency for epidermal wound-healing effect on day 3, but no statistical significance. On day 6, both the 12 and 25 μM seletinoid G treated groups showed a slightly faster wound-healing efficacy than the control group, although there was no significance.

### 2.3. Gene Expression of Proliferation and Migration in Seletinoid G-Treated HaCaT cells

Having observed that seletinoid G promoted wound healing in the epidermis, we further elucidated the gene expression level of factors known to affect keratinocyte proliferation and/or migration (two major mediators of wound healing). We treated HaCaT cells with seletinoid G for 24 h and used real-time PCR to detect the gene expression levels of keratinocyte growth factor (*KGF*), microRNA-31 (*miR-31*), keratin 1 (*KRT1*), keratin 10 (*KRT10*), *KI-67* and proliferating cell nuclear antigen (*PCNA*). As shown in Figure 4, seletinoid G significantly and dose-dependently increased the mRNA levels of *KGF*, *miR-31*, *KRT1* and *KRT10* (all stimulators of keratinocyte proliferation), whereas the mRNA expression of *PCNA* and *KI-67* was not altered under seletinoid G treatment.

### 2.4. Collagen Deposition in UVB-Irradiated Human Skin Equivalents Treated with and without Seletinoid G

A previous study showed that seletinoid G increased the expression of procollagen and reduced that of matrix metalloproteinase (MMP)-1 in skin [12]. To examine collagen deposition in our system, we used second harmonic generation (SHG) imaging to compare the auto-fluorescence signal intensity and alignment of collagen in the dermis layers of control, UVB-irradiated and UVB-irradiated/seletinoid G-treated human skin equivalents. We monitored collagen deposition by the depth (z), which was defined as the distance from the bottom of the skin tissue. As shown in Figure 5A, the collagen signal was weaker in UVB-irradiated tissues than in control tissues, whereas seletinoid G potently recovered collagen expression at all tested depths of UVB-irradiated skin equivalents. Moreover, the collagen fibrils were clearly cross-linked in UVB-irradiated samples but appeared relatively normal in UVB-irradiated/seletinoid G-treated skin equivalents. The amount of secreted MMP-1 was also increased by UVB irradiation, and this increase was suppressed by seletinoid G treatment of UVB-irradiated human skin equivalents (Figure 5B).

## 3. Discussion

Skin aging is a combination of biochemical, mechanical and environmental changes that lead to declines in the structure and function of the skin. These changes may be induced by the passage of time (chronological aging) and/or chronic exposure to solar UV irradiation (photo-aging) [17,18], and they result in functional deficits that make the skin more susceptible to injury and can delay or impair the healing process [19]. As a result, older individuals are predisposed to wound infection, trauma and the development of chronic wounds [6]. Thus, efforts to improve wound healing among older individuals can support the structure and function of aged skin and yield anti-aging effects.

Previous studies suggested that seletinoid G may be as effective as tretinoin in treating intrinsic and/or photo-aging [12,13]. However, little data were available regarding the wound-healing ability of seletinoid G. To address this gap, we examined the potential for seletinoid G to increase wound healing and then assessed its possible impact on the gene expression levels of factors known to stimulate the proliferation and/or migration of keratinocytes (e.g., growth factors, structural filaments and cell cycle mediators). We examined the following: keratinocyte growth factor (*KGF*, alternatively designated *FGF-7*), which stimulates migration of human keratinocytes [20]; microRNA-31 (*miR-31*), which has been implicated as an important regulator of keratinocyte biology and been shown to promote skin wound healing by enhancing keratinocyte proliferation and migration [21,22]; keratins (*KRT*), which impart mechanical strength to a keratinocyte and are the major structural proteins of the vertebrate epidermis [23]; and proliferating cell nuclear antigen (*PCNA*, also called cyclin) and *KI-67*, which correlate with the proliferating state in human keratinocytes and were shown to promote keratinocyte proliferation during cutaneous wound healing [24,25]. As presented in Figure 4, seletinoid G significantly increased the gene expression levels of *KGF*, *miR-31*, *KRT1* and *KRT10*, which are related to the migration of keratinocytes, but did not significantly affect the proliferation-related factors, *PCNA* or *KI-67*. These data suggest that seletinoid G may facilitate the wound-healing process by acting on the migration of keratinocytes, rather than their proliferation, although seletinoid G slightly induces cell proliferation as shown in Figure 1.

Our quantitative real-time PCR results to confirm the gene expression of keratinocyte proliferation and migration markers showed that 25 μM seletinoid G was better than 12 μM. However, other results (monolayer scratch assay in Figure 2 and three-dimensional human skin equivalent wound-healing assay in Figure 3) showed that 12 μM is more effective than 25 μM. It is assumed that this discrepancy found in the data with respect to doses results from the presence of mechanical damage to keratinocytes. Wound-healing assays were performed under mechanically induced injury conditions, while RT-PCR studies were performed under normal cell condition. In addition, we supposed that cell culture conditions such as fetal bovine serum (FBS) contents and 2D/3D culture system would affect the effective concentrations of seletinoid G. Given these factors, it is likely that the optimal concentrations of seletinoid G could be slightly changed with in vitro experimental conditions; therefore, we concluded that the effective concentration range of seletinoid G is between 12 and 25 μM in this study.

The present study did not seek to further reveal the molecular mechanism underlying the wound-healing effect of seletinoid G. Instead, we focused on using advanced technology to clearly visualize the wound-healing process in our systems. First, we used time-lapse imaging microscopy to monitor the wound-healing process in real-time. As shown in Appendix A and Figure 1, our real-time observations of healing monolayers revealed that keratinocytes moved faster in the presence of seletinoid G. Next, we utilized swept-source optical coherence tomography (SS-OCT) to evaluate the wound-healing effect of seletinoid G on an in vitro human skin equivalent. OCT is a near-infrared imaging technique that collects interference signals reflected from the specimen to reconstruct a cross-sectional image. It enables morphological structures to be visualized at up to sub-cellular resolution in a real-time and non-invasive manner. This quantitative tissue monitoring method may be used to evaluate tissue regeneration after skin injury [26,27] and is considered to be a powerful modality for imaging the function and structure of skin tissues during wound healing. We also estimated the effect of seletinoid G on collagen density in UVB-irradiated human skin equivalents by applying second harmonic generation (SHG) imaging, which can be used to provide high-resolution three-dimensional maps of collagen cross-link autofluorescence [6,28]. A previous report showed that seletinoid G increased type I procollagen and reduced matrix metalloproteinase (MMP)-1 expression in skin in vivo [12]. As reduced collagen deposition and increased cross-linking affect cell metabolism processes, such as wound healing [6,29,30], it was notable that the prior study measured collagen by Western blotting and did not provide data on the collagen cross-linking status. Here, using SHG, we found for the first time that seletinoid G could rescue the collagen level decrease, collagen fibril cross-linking and MMP-1 secretion increase triggered by UVB irradiation of human skin equivalents (Figure 5). These data suggest that seletinoid G contributes to correcting the UVB-induced alterations of the extracellular matrix (ECM), which include collagen deficiency and cross-linking.

Retinoids are popular cosmetic ingredients for anti-aging. However, they are associated with side effects such as skin problems. For this reason, dermatologists have tried to develop less irritating but comparably effective retinoids. In the previous study, synthetic retinoid seletinoid G has been proposed as a successful retinoid for anti-intrinsic/photo-aging [13]. The obtained results suggested the possibility that seletinoid G could be a potent and promising cosmetic ingredient without side effects. In the current study, our results provide important evidence supporting the idea that seletinoid G can improve skin barrier function by promoting keratinocyte migration in the epidermis to facilitate wound healing and by recovering abnormal collagen deposition in the dermis. These observations strongly supported that seletinoid G improves the whole skin tissue, although further research is needed. Collectively, we suggest that seletinoid G could be a potent candidate for development as a new-generation retinoid for anti-aging.

## 4. Materials and Methods

### 4.1. Materials

Seletinoid G, which is a retinoid analog previously designed by computer-aided molecular modeling, was synthesized as previously described [12].

### 4.2. Cell Culture and Viability Assay

The human keratinocyte cell line, HaCaT, was purchased from CLS (#300493, Cell Lines Service, Eppelheim, Germany). Normal human dermal fibroblasts, neonatal (NHDF), were purchased from Thermo Fisher Scientific (#C-004-5C, Waltham, MA, USA). HaCaT cells and NHDF were cultured in Dulbecco’s modified Eagle’s medium (DMEM; #12-604F, Lonza, Walkersville, MD, USA) containing 10% fetal bovine serum (#10082-147, Thermo Fisher Scientific), 100 U/mL potassium penicillin and 100 mg/mL streptomycin sulfate (#17-602E, Lonza) at 37 °C in a humidified 5% CO_2_ incubator. The rate of cell viability was quantified using the Cell Counting Kit-8 (CCK-8) reagent (#CK04-11, DOJINDO, Tokyo, Japan) according to the manufacturer’s instructions.

### 4.3. In Vitro Cell Scratch Assay

HaCaT cells were seeded to a 24-well plate and cultured in DMEM containing 10% FBS. After 24 h, the cells were incubated in DMEM containing 1% FBS overnight. Each monolayer was scratched with a P1000 pipet tip and then treated with or without seletinoid G in DMEM containing 1% FBS. The scratched regions (*n* = 4 per group) were recorded every hour for 48 h using automated time-lapse imaging microscopy (JuLI Stage Real-Time Cell History Recorder, NanoEnTek, South Korea). The wound-healing area (% of the initial wound area at t = 0 h) was calculated with the JuLI-STAT software (NanoEnTek, Seoul, South Korea).

### 4.4. Human Skin Equivalent Model

We purchased a human skin equivalent wound model (EpiDermFT^TM^, #EFT-400-WH, MatTek, Ashland, MA, USA) that had a circular wound in epidermis induced by a 3-mm biopsy punch. The skin equivalents were cultured in medium (#EFT-400-ASY, MatTek), and the medium was exchanged every day. On days 0, 2, 4 and 6, 25 μL of PBS containing DMSO (vehicle) or seletinoid G (12 or 25 μM) was dropped onto the surface of the wounded area. On days 3 and 6, EpiDermFT tissues (*n* = 3 per group) were fixed and subjected to OCT imaging (see below), and then paraffin embedded and sectioned at 10 μm for H&E staining.

### 4.5. Optical Coherence Tomography (OCT) for In Vitro Live Imaging

Briefly, image reconstruction was performed as follows. The light was emitted from a swept laser (Axsun Tech., Billerica, MA, USA), which had a 1310-nm center wavelength and a 110-nm tuning range. The light was propagated along the optical fiber and divided by a coupler that moved it toward the sample and reference arms. We designed a Mach–Zehnder interferometer, then the light returned from each arm through a circulator to form an interference signal. The signal was detected by a balanced amplified photodetector (Thorlabs Inc., Newton, NJ, USA) and digitized by a digitizer (AlazarTech Inc., Pointe-Claire, Canada) to create an intensity profile that reflected information on depth. The system had an axial resolution of ~7 µm and a lateral resolution of ~15 µm in the air; each cross-sectional image consisted of 1000 A-lines obtained with an acquisition speed of 20 frames/s.

### 4.6. RNA Extraction, cDNA Synthesis and Quantitative Real-Time Polymerase Chain Reaction (qRT-PCR)

Total RNA was isolated using an RNeasy Mini Kit (#74104, Qiagen, Hilden, Germany) or an RNeasy Plus Mini Kit (#74134, Qiagen) and cDNA was prepared using a SuperScript III First-Strand synthesis system kit (#51101, Invitrogen, Grand Island, NY, USA). TaqMan primer sets for the *KGF* (#Hs00940253_m1), *miR-31* (#4427975), *KI-67* (#Hs04260396_g1), *KRT1* (#Hs00196158_m1), *KRT10* (#Hs00166289_m1), *PCNA* (#Hs00427214_g1) and *RPLP0* (#4333761F) genes were purchased from Applied Biosystems (Foster City, CA, USA). qRT-PCR was performed using the TaqMan universal PCR master mix (#4304437, Applied Biosystems) and an Applied Biosystems 7500 Fast Real-time PCR system. Relative mRNA expression levels of target genes were normalized to those of the housekeeping gene, *RPLP0*, and calculated using the comparative ∆∆C_t_ method, according to the manufacturer’s instructions.

### 4.7. Second Harmonic Generation (SHG) Imaging

The optical setup for multi-photon microscopy was as described previously [31]. To visualize collagen fibrils in the dermis of our human skin equivalent, we used second harmonic generation (SHG) imaging (excitation wavelength = 810 nm / emission wavelength = 405/10 nm). Briefly, human skin equivalents (EpiDermFT^TM^, #EFT-400, MatTek) were irradiated with 30 mJ/cm^2^ of UVB (average intensity: 2.6 mW/cm^2^) using a BIO-SUN UV-H (Vilber Lourmat, Marne-la-Vallée, France), and then treated with or without seletinoid G (20 μM) for 48 h. The EpiDermFT tissues were then fixed in 4% formalin for 2 h at room temperature and washed with PBS/0.1% bovine serum albumin. The dermal (bottom) layer of each tissue was placed on a Lab-Tek II chambered coverglass (#144379, Nunc, Naperville, IL, USA) and imaged from its lower (z-depth = 0 μm) to upper regions (z-depth = 150 μm).

### 4.8. Enzyme-Linked Immunosorbent Assay (ELISA) for MMP-1

We collected the culture medium of human skin equivalents that had been exposed or not to seletinoid G for 48 h. The concentrations of MMP-1 protein in the culture medium were measured using an MMP-1 ELISA kit (#DY901, R&D Systems, Minneapolis, MN, USA) according to the manufacturer’s protocol.

### 4.9. Statistical Analysis

Data are expressed as the mean ± standard deviations (SDs), and statistical significance was analyzed by the Student’s *t*-test. *P*-values less than 0.05 were considered statistically significant.

## Figures and Tables

**Figure 1 ijms-21-03198-f001:**
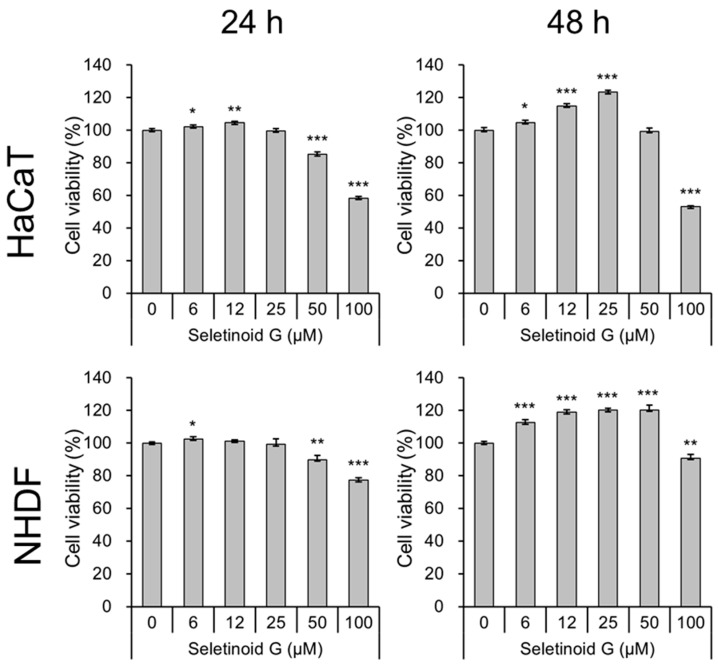
Cell viability test of seletinoid G on a human keratinocyte cell line (HaCaT) and normal human dermal fibroblasts (NHDF). The cell viability of HaCaT cells and NHDF treated with seletinoid G at different concentrations for 24 and 48 h was measured by CCK-8 assay. (* *p* < 0.05; ** *p* < 0.01; *** *p* < 0.001 vs. the untreated group).

**Figure 2 ijms-21-03198-f002:**
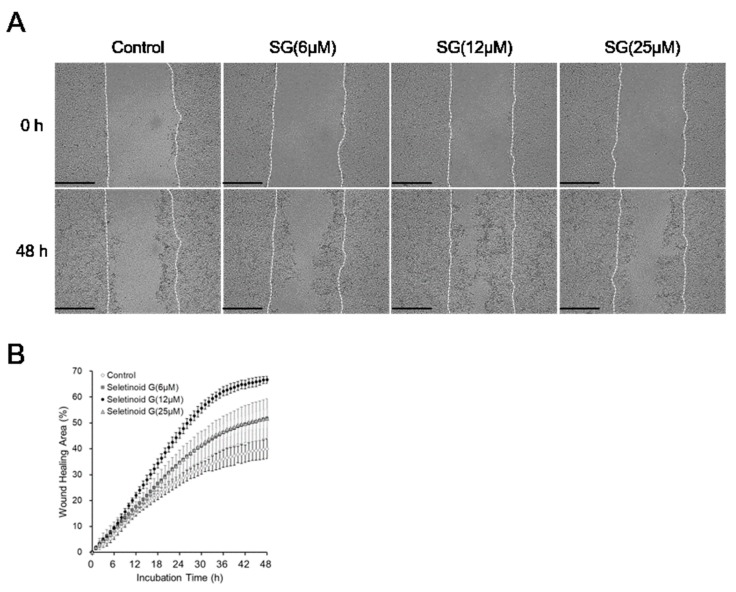
In vitro wound-healing effect of seletinoid G on wounded HaCaT keratinocyte monolayers. (**A**) HaCaT cells were line-scratched and then treated with seletinoid G (SG) at concentrations of 6, 12, and 25 μM in Dulbecco’s modified Eagle’s medium (DMEM) containing 1% fetal bovine serum (FBS) for 48 h Scale bars indicate 500 μm. (**B**) Each line-scratched area was automatically measured every hour for 48 h using time-lapse imaging microscopy.

**Figure 3 ijms-21-03198-f003:**
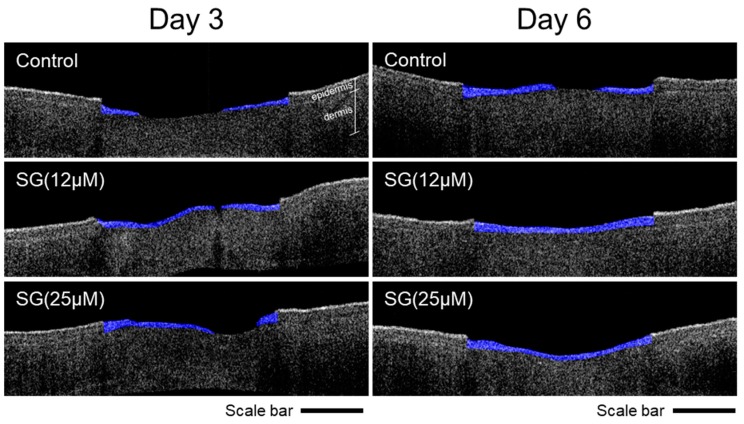
In vitro wound-healing effect of seletinoid G on a human skin equivalent wound model (MatTek, EFT-400-WH). Seletinoid G (12 or 25 μM) was applied every other day; on days 3 and 6, the skin equivalents were fixed with 4% formaldehyde (*n* = 3 per group). Three-dimensional imaging was performed using optical coherence tomography (OCT) microscopy. Blue color indicates the regenerated epidermal area. Scale bars indicate 1 mm.

**Figure 4 ijms-21-03198-f004:**
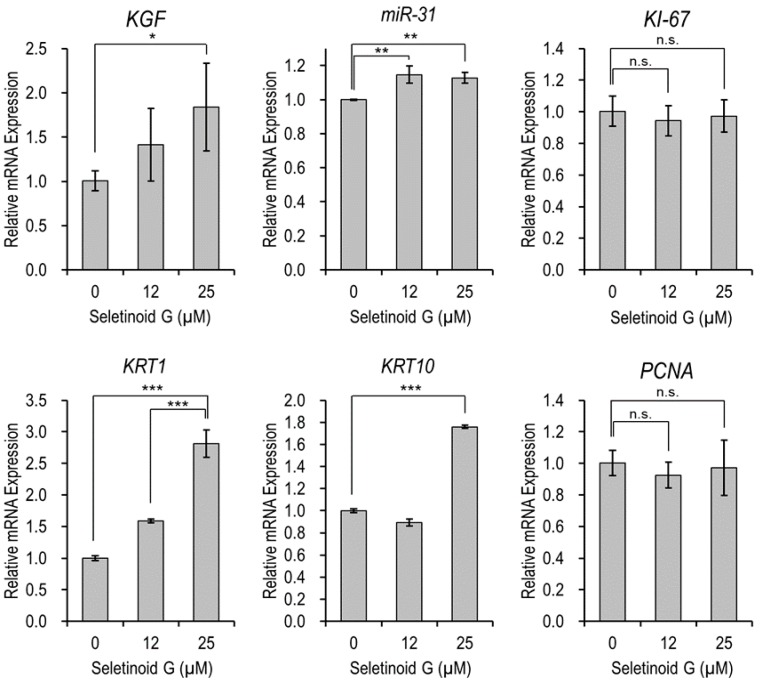
mRNA expression levels of proliferation- and migration-related genes in seletinoid G-treated HaCaT cells. The ribosomal protein lateral stalk subunit P0 (*RPLP0*) gene was used as an internal control for quantitative real-time PCR. (n.s.: not significant; * *p* < 0.05; ** *p* < 0.01; *** *p* < 0.001).

**Figure 5 ijms-21-03198-f005:**
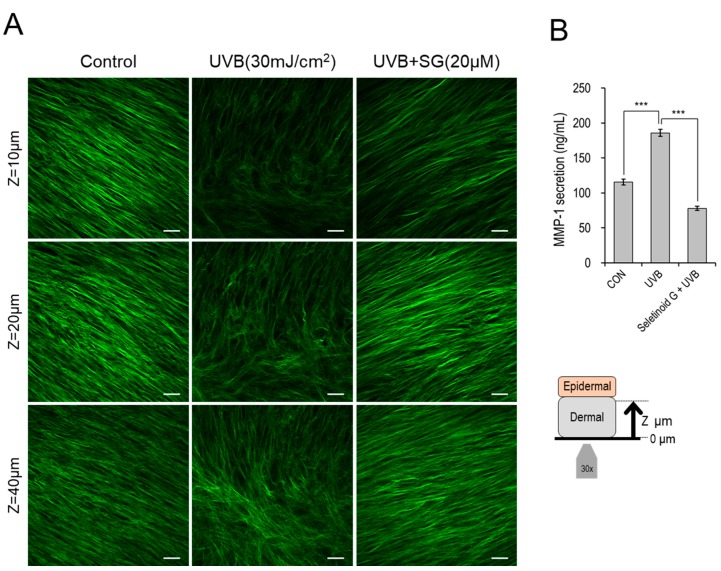
(**A**) Label-free multi-photon images of collagen fibrils (green) at different z-depths of the dermis in human skin equivalents (EFT-400) that were irradiated with UVB (30 mJ/cm^2^), irradiated and treated with SG (20 μM) in the culture medium for 48 h, or left non-irradiated and untreated (control). Scale bars indicate 40 μm. (**B**) ELISA of matrix metalloproteinase (MMP)-1 secreted to the culture medium of human skin equivalents (*n* = 3 per group). (***; *p* < 0.001).

**Table 1 ijms-21-03198-t001:** The quantification of wound-healing area, as shown in Figure 3. In a human skin equivalent wound model, we quantify wound-healing area (% of initial wound area at day 0, *n* = 3 per group) by measuring the regenerated lengths in epidermis via an Image J software from individual OCT XZ-axis images. (“*p* value” vs. control group).

After Treatment	Wound Healing Area (%)			
	Seletinoid G (μM)	Control (0)	12	25
Day 3	Mean ± SD	51.3 ± 13.5	84.7 ± 14.9	72.4 ± 23.7
	*p* value	-	0.045	0.250
Day 6	Mean ± SD	89.7 ± 12.8	100.4 ± 5.9	103.3 ± 2.7
	*p* value	-	0.2562	0.1445

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
