# Peer review of "Synthetic Retinoid Seletinoid G Improves Skin Barrier Function through Wound Healing and Collagen Realignment in Human Skin Equivalents"

_ijms, 2020, doi:10.3390/ijms21093198_

Round 1
Reviewer 1 Report
The manuscript “Synthetic retinoid seletinoid G improves skin barrier function through wound healing and collagen realignment in human skin equivalents” aims to examine the epidermal wound-healing efficacy of seletinoid G on HaCaT keratinocytes and skin tissues. This manuscript is good as for contents, methodology and brings something new to the scientific literature. However before publication, some changes are still needed.
First of all language editing is suggested to improve its presentation maybe consulting an English mother tongue during the revision of the manuscript.
Specific comments:
- Please correct the abstract because the maximum number of words is 200.
- Check how is written ml throughout all the text of the manuscript and correct it with mL.
- I have some doubts about figure 1, specifically for the part of the HaCaT cells both for the 24h and 48h treatment. The authors are asked to check the statistics, because in my opinion it is incorrect. For example, regarding 24h, do the 50 and 100 µM treatments belong to the same statistical group? In the 48 h the treatments of 25 µM and 100 µM belong to the same group? Please check all the statistic and if necessary correct it. I have some doubt also for figure 4, in the graph for Kl-67, why there isn’t statistic? Also if are all the same, maybe the authors can put statistic symbols like for genes KGF or KRT10.
- Please check the abbreviations in the text and add them to the abbreviations list.
- I suggest the authors to enrich the conclusion. I advise them to read and cite the following paper: DOI: 10.3390/nu9060605
Reviewer 2 Report
The manuscript by Lee et al. describes the use of Seletinoid G for wound healing. Seletinod G has been used as anti-aging and to repair altered connective tissue in old skin. The repurposing of Selectinoid-G for wound healing seems interesting and a novel idea. The authors tested various doses of Seletinod G in standard wound healing assays using human keratinocytes cell line HaCaT. Please find below the concerns and suggestions to improve the manuscripts:
1) Cell viability data (Figure 1) suggested dose up to 25uM safe and while cell migration suggested a 12uM more effective than the rest of the groups? What's the rationale for this finding?
2) In vitro wound models suggest day 3 12uM have better healing although by day 6 both 12uM and 25uM heal the same. Is 12uM more than enough for a healing effect and should be discussed. Also, the authors should quantify the healing response.
3) RT-PCR studies to confirm the gene expression of various keratinocyte proliferation and migration markers are shown to be dose-dependent and do not support the previous data of 12uM being superior.
4) Why did the authors use a 20uM dose for the collagen study (Figure 4)? It should be explained.
5) The discrepancy found in the data with respect to doses should be clearly discussed in the manuscript.
Round 2
Reviewer 1 Report
The manuscript was significantly improved by the authors, who followed the reviewers' suggestions.
Reviewer 2 Report
No more concerns. The authors have satisfactorily resolved raised concerns regarding dosing and some data.